# Evaluating the effects of embedded self-massage practice on strength performance: A randomized crossover pilot trial

Yann Kerautret[1,2], Aymeric Guillot[1,3], Franck Di Rienzo[1]*

1 Laboratoire Interuniversitaire de Biologie de la Motricité EA 7424, Université Claude Bernard Lyon 1, Univ Lyon, Villeurbanne Cedex, France, 2 Capsix Robotics, Lyon, France, 3 Institut Universitaire de France, Paris, France

* franck.di-rienzo@univ-lyon1.fr

**Data Availability Statement:** All relevant data are within the paper and its Supporting Information files.

**Funding:** The authors received no specific funding for this work. CAPSIX robotics, a commercial company, provided support in the form of salaries

## Abstract

### Background

Self-administered massage interventions with a roller massager are commonly used as part of warm-ups and post-workout recovery routines. There is yet no clear consensus regarding the practical guidelines for efficient embedded interventions.

### Objectives

The present randomized crossover pilot trial aimed at examining the effects of a rolling intervention with a roller massager embedded within the rests periods of a resistance training protocol. The rolling intervention targeted quadriceps muscles.

### Setting

Participants (n = 14) performed two resistance training protocols expected to elicit momentary muscle failure. The protocol consisted in 10 sets of 10 rest-pause repetitions of back squats, with a poundage set up at 50% of the maximal one-repetition. Two min were allocated to recovery between sets. During the recovery periods, participants completed a rolling routine with a roller massager for 60 s (Roller-massager), or underwent passive recovery (Control). The total workload, concentric power, thigh circumference rate of perceived exertion (RPE) and delayed onset of muscle soreness (DOMS) from 24 h to 120 h after completion of the protocol were the dependent variables.

### Results

Roller-massager was associated with a reduction in total workload (-11.6%), concentric power (-5.1%) and an increase in perceived exertion compared to Control (p < 0.05). Roller-massager was also associated with reduced thigh circumference after the resistance training protocol, indicating reduced muscle swelling, and reduced DOMS 24 h to 120 h post-workout (p < 0.001).

for one of the authors (YK), but did not have any additional role in the study design, data collection and analysis, decision to publish, or preparation of the manuscript. The specific roles of all co-authors are articulated in the 'author contributions' section.

**Competing interests:** The authors declare no conflict of interest. The commercial affiliation to CAPSIX Robotics does not alter authors' adherence to all PLOS ONE policies on sharing data and materials. CAPSIX Robotics is not a company promoting self-myofascial release or conditioning products.

## Conclusion

These findings support that embedded rolling with a roller massager hinders performance and increases effort perception. Embedded interventions may not be suitable during conditioning periods designed to maximize training intensity.

## Introduction

Rolling techniques consist of self-stimulation of the soft tissues with a foam roller, a roller massager, sticks or balls of varying sizes, surfaces and densities [1–5]. SMR represents a simple and cost-effective conditioning approach, increasingly used as part of warm-up and post-workout recovery routines and, occasionally, embedded within training sessions [6–9]. Although physiological processes underlying the benefits of SMR remain poorly understood, SMR interventions have practical relevance in actual training contexts compared to static stretching [1, 10–14]. Pre-workout SMR interventions were associated with increased range of motion (ROM) without hindering forthcoming athletic performances such as jump height or performance across repeated sprints [15–17]. Post-workout, SMR reduced exercise-induced muscle damage indexed from the delayed-onset muscle soreness (DOMS) [18]. However, there is no scientific consensus regarding optimal practice guidelines for embedded SMR and its acute effects on strength performance [6, 8, 9, 12, 14].

Over the last decade, SMR has become increasingly popular in fitness communities. Cross-Fit® athletes (CrossFit, Inc, Washingtion, DC, USA), for instance, regularly use SMR as part of their training routines to improve joint mobility [11, 19]. Athletes are in control of the pressure, speed, frequency and other fine adjustments when performing SMR routines, which might account for their popularity. While SMR is frequently used as part of the warm-up or post-workout recovery routines, only few studies evaluated its efficacy when embedded to the actual course of physical training, i.e. performed during the recovery periods of a training session. Monteiro and Neto [6] reported a deleterious effect of SMR with a grid foam roller on strength performance and a corollary increased in the rate of perceived exertion (RPE). The embedded SMR practice with a grid foam roller of agonist muscles impaired performance of both agonist and antagonist muscles in a resistance training paradigm, and these effects were more pronounced after 120 s compared to 60 s of SMR [8, 9, 20]. These results challenge the assumption that embedded SMR practice could be beneficial to strength performance in resistance training paradigms. This is somehow surprising considering that the mechanical and neurophysiological effects of SMR could suggest beneficial effects on strength performance. SMR has been shown to increase muscle compliance to the effort by affecting the mechanical properties of contractile tissues, and promote lactate clearance [6, 8, 9]. Monteiro et al. [9] postulated on an endogenous opioid response. This response would bias effort perception by attenuating afferent feedback to the central nervous system from the somatic effectors. Increased power output due to the downregulation of noxious afferent feedback would delay fatigability. The authors also hypothesized that SMR yielded additional cognitive and physical demands that might increase fatigability during the resistance training protocol. Due to the scarcity of available experimental data and contradictory findings with regards to the benefits of SMR decoupled or embedded within the training practice, further investigation of SMR interventions in strength training paradigms is required. Here, we administered a SMR intervention with a roller massager, targeting quadriceps muscles, during the inter-set periods of a resistance training protocol in advanced CrossFit® athletes. Based on Monteiro and Neto [6],

we hypothesized that embedded SMR with a roller massager would reduce the strength performance. We were also interested in the influence of embedded SMR with a roller massage on the perceived exertion and DOMS up to 120 h after the completion of the resistance training protocol. Overall, we aimed at extending current knowledge regarding embedded SMR intervention in applied training settings.

## Materials and methods

### Participants

Fourteen healthy physically active (8 males and 6 females, mean age 25.9 ± 2.6) adults volunteered to participate in the present randomized crossover pilot trial (Table 1). Recruitment strategies consisted in poster advertising and personal contact between Crossfit® coaches and practitioners. Inclusion criteria were the regular practice of SMR with a roller massager (10 min routines, 2–3 times per week, over the past 6 months) and the regular practice of Crossfit®. We requested a minimum experience of one year in weightlifting practice (2–3 training sessions of > 45 min per week), ensuring steady baseline performances in back squat practice. Our sample was thus representative of amateur CrossFit athletes dedicated to a regular practice on a weekly basis. All participants were right-legged, as revealed by their self-reported leg dominance and scores on the Waterloo Footedness Questionnaire-Revised [21]. Participants were free from medical conditions, including functional limitations which could have confounded results. The experiment was approved by the local ethics committee (Human subject research —human participants; Comité de Protection des Personnes Ouest VI; CPP—Ouest 6—CPP 1223 HPS2). A written form of consent was obtained (2019-A01732-55), in accordance with the ethical standards laid down in the Declaration of Helsinki and its later amendments [22].

### Experimental design

The experiment was scheduled within a span of 30 consecutive days. The experimental design involved a familiarization session, followed by two experimental sessions separated by 10 consecutive days. Participants were instructed to not engage in any strenuous resistance training exercises between the familiarization and the experimental sessions. Since few consecutive days without proper stimulation could downregulate strength performance [23], participants were allowed to maintain their regular practice of weightlifting at maintenance levels. Also, the use of a crossover design controlling for experimental sessions order (using block randomization, see below) was expected to control for potential changes in strength performance between the two experimental sessions [23].

**Familiarization session.** The familiarization session took place 7 days before the first experimental session, and was used to collect demographics (Table 1). Participants completed

**Table 1. Participants' characteristics by sex (M ± SD).**

| Characteristics | Age (years) | Height (cm) | Mass (kg) | Body mass index (m2/kg) | Weightlifting experience (months) | 1RM back squat (kg) |
|---|---|---|---|---|---|---|
| Males | 26.5 ± 1.7 | 177.8 ± 5.8 | 76.3 ± 10.2 | 24.0 ± 1.8 | 17.8 ± 7.7 | 139.8 ± 23.6 |
| | (range 24–29) | (range 170–187) | (range 58–95) | (range 20.1–27.2) | (range12-36) | (range 110–178) |
| Females | 25.2 ± 2.6 | 167.7 ± 4.6 | 61.8 ± 5.9 | 21.9 ± 1.2 | 15.7 ± 4.5 | 99.8 ± 10.2 |
| | (range 23–30) | (range 162–175) | (range 53–68) | (range 20.2–23.5) | (range 12–24) | (range 90–114) |
| Group difference (1-way ANOVA) | $F_{(1, 12)} = 1.38$, $p = 0.026$ | $F_{(1, 12)} = 12.42$, $p < 0.01$ | $F_{(1, 12)} = 9.49$, $p < 0.01$ | $F_{(1, 12)} = 5.36$, $p < 0.05$ | $F_{(1, 12)} = 0.34$, $p = 0.57$ | $F_{(1, 12)} = 14.80$, $p < 0.01$ |

There were no adverse events and no subjects withdrew during data collection.

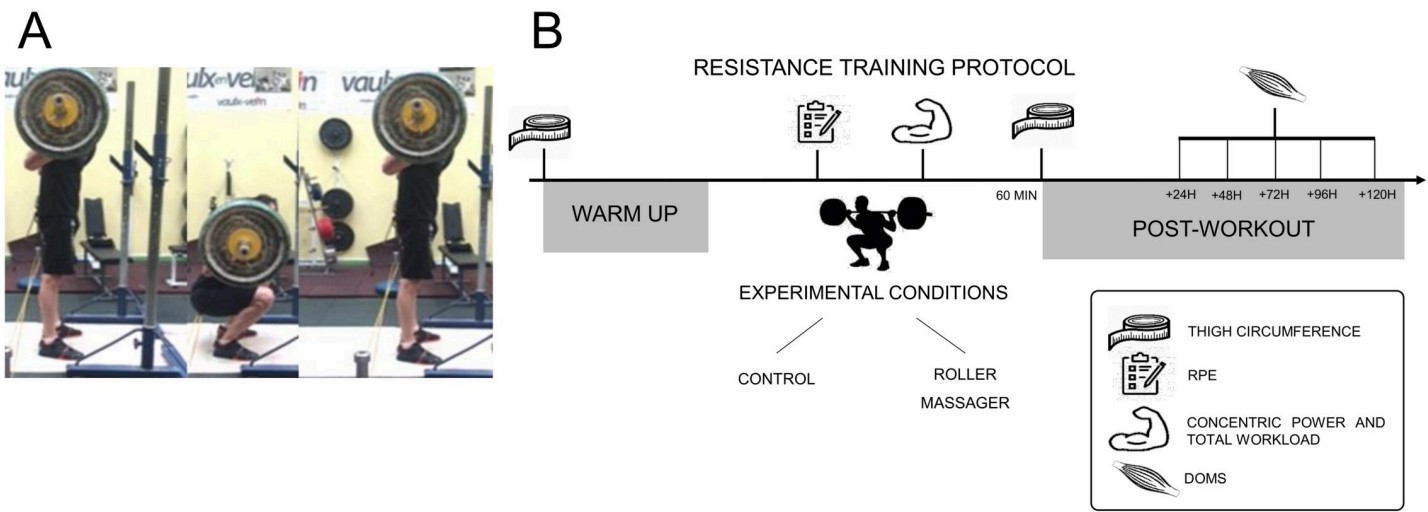

**Fig 1. Back squat movement.** A. Execution stages of the back squat. B. Flowchart summarizing the experimental protocol.

a warm-up routine consisting in 5 minutes of self-paced skipping ropes, followed by 5 minutes of dynamic stretching aiming at improving joint mobility. Then, back squat practice was administered. First, we controlled feet's position with an empty weightlifting bar, i.e. feet distance and foot opening, and the distance between the buttocks and the ground in the deep squat position (after the eccentric phase). Each participant was asked to perform a maximum deep squat with an empty barbell, respecting the technical rules of the International Powerlifting Federation "*the lifter must bend the knees and lower the body until the top surface of the legs at the hip joint is lower than the top of the knees*" [24]. Once the depth was reached, the distance was materialized with an elastic band, parallel to the ground to guarantee the interindividual reproducibility of each repetition over the two sessions. Participants then completed back squats with incremental poundage, up to their maximal 3–5 repetition (3–5 RM). The 3–5 RM was reported to Berger's table to provide an indirect estimate of the 1RM [25, 26]. Participants were required to achieve contact between the buttocks and the elastic band for each squat repetition (Fig 1A).

The last part of the familiarization consisted in SMR practice with a high-density roller massager. We used a commercial roller massager made from hard plastic (MSGT01, Physioroom, Halifax, England, 45 cm). The experimenter first demonstrated and verbalized instructions regarding tempo, level of pressure and SMR duration for each massage zone. The routine focused quadriceps muscles. Participants were instructed to perform the routine with the roller massager so that the perceived pain would not exceed a 7 out of 10 threshold on the Numeric Pain Scale (0: "No pain", 10: "Most pain") [4, 27]. This subjective control is widely used in SMR experiments [28–30].

**Experimental sessions.** For the first experimental session, participants were randomly allocated to one of the following two experimental conditions over an inclusion period of 4 weeks between March to May 2016. Experimental sessions were scheduled between 4 and 6 pm. All subjects underwent pretest measures, followed by the intervention, and then immediate posttest measures. The investigator was blinded to the condition assignment but remained the same throughout the study to ensure reproducibility of the measures. No feedback was provide to the participants until after completion of the design.

*Resistance training protocol.* After a standardized warm-up, participants engaged in 10 sets of 10 back squat repetitions with a poundage set up at 50% of the individual 1 RM estimate (Fig 1B). 2 min were allocated to recovery between sets. Back squats were performed using a tempo of "2111" (2 s allocated to the eccentric phase, 1 s of pause in deep squat position, 1 s for the concentric phase, and 1 s of pause between repetitions). The rest-pause tempo represents a dramatic increase in task difficulty and likelihood to elicit momentary muscle failure. An auditory device cued the tempo. When the participant was no longer able to follow the tempo or reached momentary muscle failure, the experimenter stopped the protocol. This protocol aimed at reproducing training patterns targeting muscle hypertrophy. Approaching momentary muscle failure is a reliable method to increase motor units recruitment, elicit myofilament damage and prompt anabolic endocrine responses necessary for muscle growth [31].

*Experimental conditions.* Experimental conditions were administered during the inter-set periods allocated to recovery. During CONTROL, participants were asked to remain passive in a comfortable seated position. During ROLLER MASSAGER, participants kept one leg semi-stretched to relax quadriceps muscles. Then, they completed the SMR routine, for both legs alternately. Participants foam rolled the quadriceps in the proximal-distal axis, (from the anterior superior iliac spine to the top of patella), at a pace of 15 beats per minute. Each zone of the quadriceps, i.e. medial, lateral and external, was massaged for 10 s. This represents an effective SMR duration of 30 s. The standards for the present SMR routine matched those reported in previous experiments [32–36].

## Dependent variables

**Behavioural variables.** *Total workload.* The total workload corresponded to the total poundage lifted by the participant throughout the entire protocol. The number of repetitions until momentary muscle failure, from both complete and incomplete sets were collected using a hand-held mechanical counter. The total workload was then calculated based on the following formula:

$$Total\ workload = Total\ repetition\ number * Workload\ (50\%\ of\ 1RM\ estimate)$$

To control the correct execution of the back squats, i.e. contact with the elastic band for each rest-pause repetition, a Go Pro camera was installed on a tripod (Hero 4 Black, GoPro, San Mateo, United States, 720 pixels, 120 frames per second). Video recordings of the repetitions were analyzed offline using Kinovea 0.8.15 (Kinovea project, France).

*Concentric power.* To evaluate the average concentric power an accelerometer was attached to the weightlifting bar using velcro (Myotest PRO, Sion, Switzerland, 250 Hz). The concentric power developed for each repetition was obtained using Myotest PRO algorithms and averaged across sets. The experimenter ensured that the device remained vertical during the sets to prevent artifacts.

*Thigh circumference.* Thighs circumference was collected before and after the resistance training protocol using a meter ruler 15 cm above the top of the patella, avoiding deformation of the skin. Measuring thigh circumference 15 cm above the patella represents a well-acknowledged methodological standard [37, 38]. To replicate the assessment, participants were marked with permanent marker. All evaluations were determined to the nearest 0.1 cm.

**Psychometric variables.** *Rate of perceived exertion.* After each set, participants were reported their RPE on a Borg's category-ratio scale (CR-10; 0: "Nothing at all", 10: "Extremely strong") [39, 40].

*Delayed onset of muscle soreness.* DOMS measures involved five visual analog scales of 10 cm (0: "No pain"; 10: "Very severe pain") [41], corresponding to five muscle groups (i.e.

quadriceps, hamstrings, adductors, gluteus, erector spinae), for both legs. DOMS measures were collected before and after the resistance training protocol, and 24 h, 48 h, 72 h, 96 h and 120 h after its completion.

## Statistical analyses

**Block randomization.** We used R [42] and the package *blockrand* to achieve participant's conditions (block randomization) [43]. We assigned the order of the two experimental conditions (i.e. ROLLER MASSAGER first or CONTROL first). *Blockrand* allows block randomization with random block size selection to prevent randomization bias [44]. The randomization procedure was completed in two block for males (n = 6 and n = 2, respectively), and one block for women (n = 6).

**Data analysis.** We first tested the effect of CONDITION (Roller massager, Control) on total workload using a paired t-test. We then ran a series of linear mixed effects analyses using *nlme* [45]. Visual inspection of residual plots did not reveal any obvious deviations from homoscedasticity or normality [46]. For the analysis of the total workload, number of sets and number of repetitions per set, we included the fixed effect of CONDITION (ROLLER MASSAGER, CONTROL). For the concentric power and RPE scores we added the fixed effect of and SET (numeric regressor, 1–10), with interaction term. For thigh circumference, we entered CONDITION, LATERALITY (LEFT, RIGHT) and TEST (PRETEST, POSTTEST), with interaction terms. For DOMS ratings, we entered the interaction between CONDITION and TEST, MUSCLE (QUADRICEPS, HAMSTRINGS, ADDUCTORS, GLUTEUS, ERECTOR SPINAE) and TIME (numeric regressor, i.e. time delay in hour after completion of the resistance training protocol) without interaction terms. The statistical significance threshold was set up for a type 1 error rate of 5%. As effect sizes, we calculated partial coefficients of determination ($R_P^2$) using an ad-hoc procedure for linear mixed effects models implemented in the *effect size* package [47]. Main effects and interactions were investigated using general linear hypotheses testing of planned contrasts from the *multcomp* package. We applied Holm's sequential corrections to control the false discovery rate [48].

**Power/Sample size considerations.** Considering the pilot nature of the study, we did not run a priori power calculation. Implementing a counterbalanced cross-over design with block randomization was expected to increase the statistical power compared to between-subject group designs. We thus ran a posteriori power ($p_{1-β}$) calculations using the *pwr* package [49] for statistically significant main and interaction effects revealed by the linear mixed effects analysis.

## Results

### Behavioral data

**Momentary muscle failure.** All participants reached momentary muscle failure during ROLLER MASSAGER. Ten participants reached momentary muscle failure during CONTROL. CONDITION affected the total workload (F(1, 13) = 13.68, $R_p^2$ = 0.51, p < 0.01, $p_{1-β}$ = 0.72).

**Total workload.** Participants achieved a total workload of 4549.21 kg ± 1123.88 during ROLLER MASSAGER, and 5146.29 ± 1046.53 during CONTROL. CONDITION also affected the number of set (F(1, 13) = 9.75, $R_p^2$ = 0.43, p < 0.01, $p_{1-β}$ = 0.65). Participants achieved 8.0 ± 2.07 (M ± SD) sets during ROLLER MASSAGER and 8.85 ± 1.75 sets during CONTROL. However, there was no CONDITION effect for the average number of repetitions per set (F(1, 13) = 1.59, p = 0.22), with respectively 9.59 ± 0.21 and 9.71 ± 0.34 repetitions per set for the ROLLER MASSAGER and CONTROL conditions.

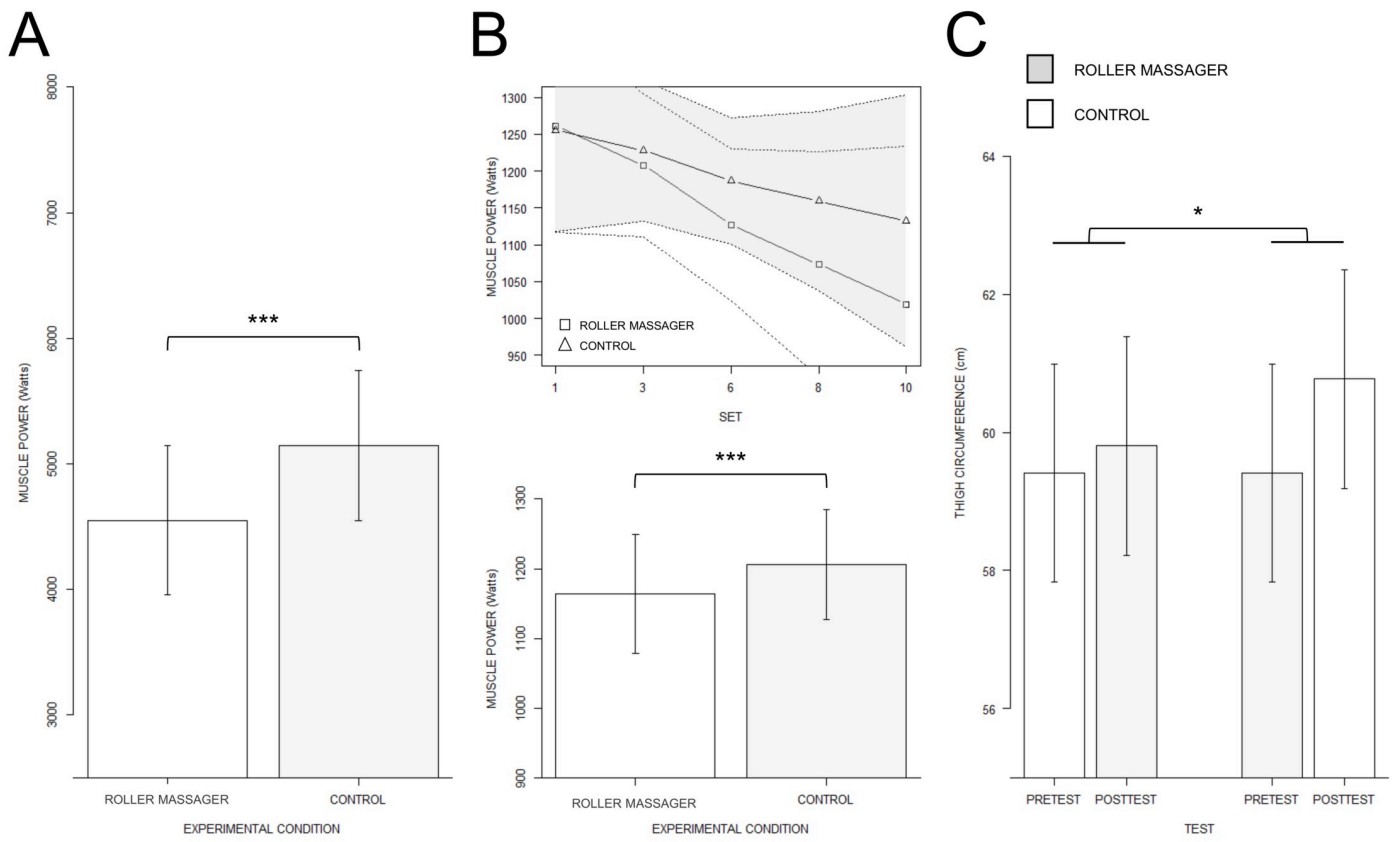

**Fig 2. Mean muscle power and thigh circumference.** A. Barplot depicting fitted estimates for the main effect of EXPERIMENTAL CONDITION on concentric power. B. Regression slopes depicting the SET by CONDITION interaction revealed by the linear mixed effects analysis carried on muscle power. C. Barplot display of the TEST by CONDITION interaction effect on thigh circumference. *** $p < 0.001$, * $p < 0.01$.

**Concentric power.** The linear mixed effect analysis carried on concentric power data revealed no CONDITION by SET interaction ($p > 0.05$). However, there was a main effect of CONDITION ($F(1,191) = 8.09$, $p < 0.01$, $R_p^2 = 0.05$, $p_{1-\beta} = 0.87$) and SET ($F(1, 191) = 35.75$, $p < 0.001$, $Rp2 = 0.16$, $p_{1-\beta} > 0.95$). During CONTROL (1106.45 W ± 277.54), concentric power values were higher compared to those recorded during ROLLER MASSAGER (1049.71 W ± 240.46) (fitted difference: 66.57 ± 19.29, $p < 0.001$, see Fig 2A). SET values negatively affected concentric power output irrespective of the experimental condition (fitted estimate: -23.02 W ± 3.88, $p < 0.001$, see Fig 2B).

**Thigh circumference.** There was a CONDITION by TEST interaction effect on thigh circumference ($F(1,91) = 33.63$, $p < 0.001$, $Rp^2 = 0.27$, $p_{1-\beta} > 0.95$), while all other interactions did not reach the statistical significance threshold (all $p > 0.05$). Thigh circumference was also affected by the main effects of CONDITION ($F(1, 91) = 33.63$, $p < 0.001$, $Rp^2 = 0.27$, $p_{1-\beta} > 0.92$), TEST ($F(1, 91) = 110.40$, $p < 0.001$, $Rp^2 = 0.55$, $p_{1-\beta} > 0.95$) and LATERALITY ($F(1, 91) = 9.10$, $p < 0.01$, $Rp^2 = 0.03$, $p_{1-\beta} = 0.40$). Post-hoc investigations revealed that the PRETEST (59.41 cm ± 4.09) to POSTTEST (60.77 cm ± 4.19) difference in thigh circumference during the CONTROL condition was greater than the PRETEST (59.41 cm ± 4.09) between POSTTEST (59.80 cm ± 4.20) difference recorded in the ROLLER MASSAGER condition (fitted difference: 0.39 cm ± 0.20, $p = 0.05$, see Fig 2C). Also, thigh circumference was affected by the main effect of

LATERALITY, with lower values for the LEFT thigh (59.72 cm ± 4.23) compared to the RIGHT thigh (59.97 cm ± 4.05) (fitted difference: -0.25 cm ± 0.08, $p < 0.01$).

## Analysis of the psychometric data

**RPE data.** The CONDITION by SET interaction failed to reach the statistical significance threshold ($p > 0.05$). RPE scores were however affected by the main effects of CONDITION ($F_{(1, 219)} = 4.59$, $p < 0.05$, $Rp^2 = 0.03$, $p_{1-\beta} = 0.72$) and SET ($F_{(1, 219)} = 1016.06$, $p < 0.001$, $Rp^2 = 0.82$, $p_{1-\beta} > 0.95$). Post-hoc investigations revealed that RPE scores during the CONTROL ($5.85 \pm 2.60$) condition were lower compared to that observed in the ROLLER MASSAGER ($6.07 \pm 2.60$) condition (fitted difference: $-0.63 \pm 0.14$, $p < 0.001$). In both conditions, an increase in the RPE along with SETS repetition (fitted estimate: $0.84 \pm 0.03$, $p < 0.001$).

**DOMS data.** DOMS self-reports on the visual scales were affected by the two-way interaction between CONDITION and TEST ($F_{(1, 813)} = 6.15$, $p < 0.01$, $Rp^2 = 0.02$, $p_{1-\beta} > 0.95$). We also found a CONDITION by TIME interaction ($F_{(1, 813)} = 15.13$, $p < 0.001$, $Rp^2 = 0.02$, $p_{1-\beta} > 0.95$), while the CONDITION by MUSCLE interaction failed to reach the statistical significance threshold ($F_{(4, 813)} = 2.36$, $p = 0.06$). Eventually, DOMS were affected by the main effects of CONDITION ($F_{(1, 813)} = 30.73$, $p < 0.001$, $Rp^2 = 0.04$, $p_{1-\beta} > 0.95$), TEST ($F_{(1, 813)} = 100.00$, $p < 0.001$, $Rp^2 = 0.11$, $p_{1-\beta} > 0.95$), MUSCLE ($F_{(4, 813)} = 22.86$, $p < 0.001$, $Rp^2 = 0.10$, $p_{1-\beta} > 0.95$) and TIME ($F_{(1, 813)} = 353.03$, $p < 0.001$, $Rp^2 = 0.30$, $p_{1-\beta} > 0.95$, see Fig 3A–3C). Post-hoc analyses revealed that DOMS ratings difference between the PRETEST ($0.00 \pm 0.00$) and the POSTTEST ($1.68 \pm 2.31$) during CONTROL was higher compared to the difference between the PRETEST ($0.00 \pm 0.00$) and the POSTTEST ($1.01 \pm 1.89$) in ROLLER MASSAGER (fitted difference: $1.55 \pm 0.35$, $p < 0.001$). Post-hoc analyses also revealed that the difference between CONTROL ($0.89 \pm 1.76$) and ROLLER MASSAGER ($0.79 \pm 1.82$) (fitted difference: $0.8 \pm 0.22$, $p > 0.001$) reported for the ERECTOR SPINAE was marginally lower than the difference between CONTROL ($1.76 \pm 2.39$) and ROLLER MASSAGER ($0.71 \pm 1.59$) (fitted difference: $-0.87 \pm 0.22$, $p < 0.01$) reported in the ADDUCTORS (fitted difference: $0.86 \pm 0.22$, $p < 0.01$). Finally, we found that TIME had a greater negative relationship on DOMS ratings during CONTROL (fitted estimate: $-0.03$ a.u./h $\pm 0.00$) compared to ROLLER MASSAGER (fitted estimate: $-0.02$ a.u./h $\pm 0.00$) (fitted difference: $0.01 \pm 0.00$, $p < 0.001$).

## Discussion

The aim of this pilot trial was to investigate whether a SMR routine with a roller massager administered during inter-sets periods of a resistance training protocol improved behavioral and psychometric indexes of strength performance. The total workload at the session level was reduced by 13.1% under the SMR condition. We also found increased perceived exertion and decreased concentric power by 5.4% per set compared to the control condition. Embedded SMR thus appeared counterproductive for athletes during preparation periods where intensity is crucial to achieve peak performance. Indeed, compared to a passive recovery condition, embedded SMR appeared to induce greater fatigability. While we also observed that embedded SMR practice was associated with reduced muscle swelling and soreness up to 120 h after completion of the resistance training protocol, this occurred at the expense of the training workload. These results overall corroborate past experiments underlining the detrimental effect of embedded SMR practice on strength performance [8, 9]. Noteworthy, it is well-established that training effects such as somatic build and recovery capacities and other factors differ between men and women [50, 51]. Here, our pilot design did not control for the effect of sex. Indeed, our sample involved both males and females. While a main effect of sex can be expected in terms of training effects, little evidence suggests at this point potential interactions

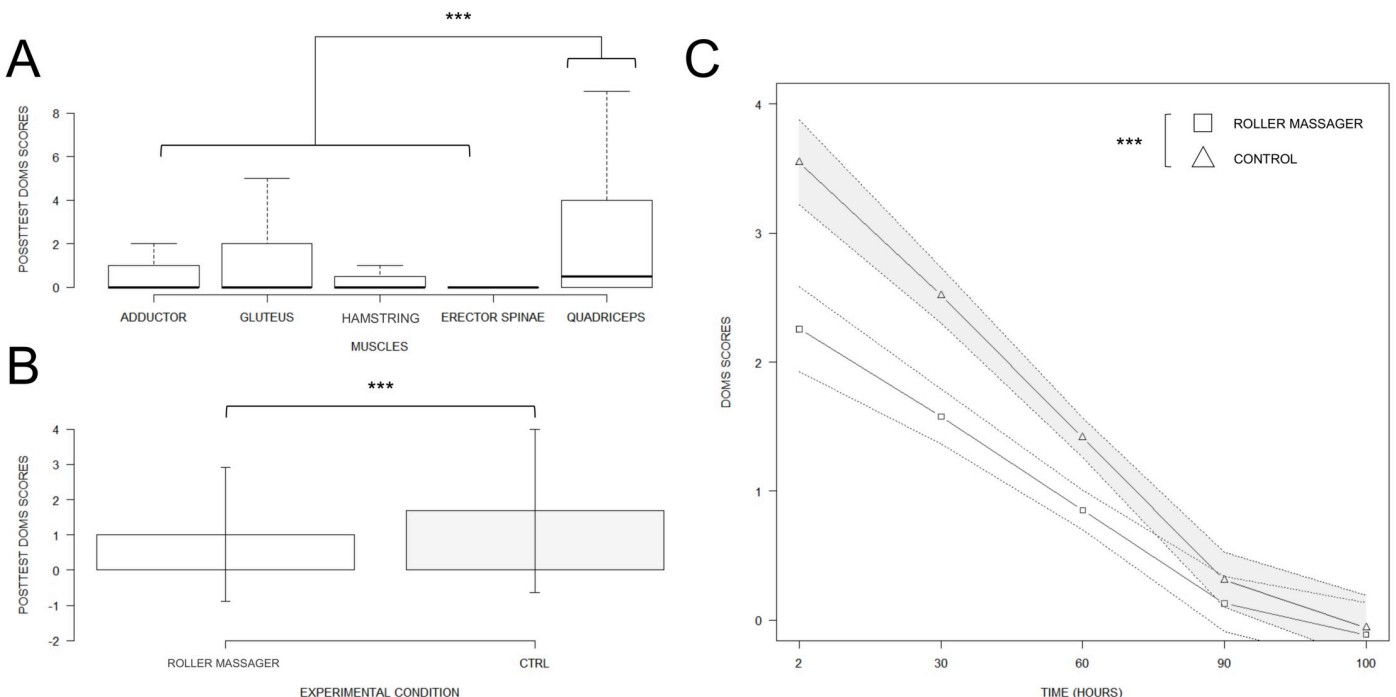

**Fig 3. DOMS intensity by muscle / condition, and its evolution over time.** A. Boxplot of general DOMS ratings. B. Barplot display of the main CONDITION effect on DOMS. C. Regression slopes attesting the TIME by CONDITION interaction effect on DOMS scores revealed by the linear mixed effects analysis. *** p < 0.001.

with the effect of massage intervention [13, 52, 53]. Addressing these issue certainly goes beyond the intent of this pilot trial, and would require methodological designs involving larger samples of participants.

The control condition consisted of passive recovery. This reproduced actual training conditions where athletes usually remain passive between the sets of their strength training sessions, hence providing ecological validity [54]. SMR yielded reduced total workload, concentric power and increased perceived exertion at the session level. This finding corroborates earlier reports of adverse effects of embedded SMR on strength performance in resistance training paradigms [6, 8, 9, 20]. Also, embedded SMR does not enhance the subjective perception of recovery [20]. After a series of knee extensions up to momentary muscle failure, 60 s or 120 s of SMR interventions on the quadriceps appeared counterproductive to restore strength performance [8]. Similar results were obtained when massaging the hamstring [6, 9]. Further, there was a dose-response relationship between SMR duration and the deleterious effects on performance [6, 8, 9], since the total workload decreased along with time spent on the SMR routine [8, 9]. While the use of SMR as part of warm-up or recovery routines does not alter muscle contractile properties [55, 56], this does not seem to apply to embedded interventions [28, 30, 57, 58]. The decreased physical performance along with increased perceived exertion under the SMR condition indicate greater fatigability [8]. SMR is a self-administered technique requiring additional cognitive and physical resources compared to passive recovery. This could emphasize fatigability elicited by the resistance training protocol, and/or interfere with recovery processes [59]. SMR has been shown to amplify structural damage to the muscles, e.g. micro-damage to the muscles altering their short-term contractile properties [60–63]. Compared to manual therapy, where pressure values are up to five-fold weaker [61, 64], SMR with a

roller massager can induce additional strain on the muscle tissues compounding with that elicited by the resistance training protocol [60, 61, 65]. It must be objected that pressure above physiological thresholds are required to induce mechanical changes, as emphasized in mathematical models carried on manual massage therapy interventions and experiments where the pressures of roller massaging were controlled [65, 66].

Thigh circumference was reduced during the posttest under the SMR condition. Muscle swelling increase was 0.97 cm (+348.71%) higher during the control condition. This might reflect the reduced total workload under the SMR condition. An important limitation is that we did not consider the measurement error for thigh circumference measures. There is thus a risk for a lack of meaningful difference in spite of statistical differences (see [8]). Indeed, the differences between pretest and posttest measures remained below the 5% range. While not surprising at the single-session level, this remains below the 5% threshold suggested as rule of thumb for the measurement error of waist circumference [67]. As a counterpoint, thigh circumference measurements were conducted by the same experimenter. This indicates that the distribution of the error across tests and experimental conditions remained homogenous. Unilateral and systematic deviations within the measurement error range would have been required to bias the results pattern. While this cannot be ruled out, the pattern of results appeared congruent with that revealed by the other dependent variables. More speculatively, the pressure levels exerted during the embedded SMR could have affected blood circulation, hence limiting muscle swelling [68–71]. By contrast, SMR post-workout did not improve muscle swelling or joint mobility or blood inflammatory markers 24 h and 48 h after a high-volume resistance training protocol in a recent experiment (10 sets of 10 repetition of back squat at 60% of the 1RM) [71]. There is also evidence supporting that the mechanical pressure elicited by the roller massager exerts blood and lymphatic drainage [72–76]. Past reports also indicate improvements in blood circulation after SMR [72–76], with improved oxygen saturation, endothelial function and lactate clearance [76–78]. While reduced muscle swelling may primarily account here for the reduced total workload under the SMR condition, we cannot rule out a possible effect on blood and lymphatic circulation. Addressing this hypothesis would require controlling the workload between SMR and control conditions.

DOMS scores were null during the pretests, indicating the lack of muscle damage, and therefore a complete recovery from one experimental session to another [79]. Muscle soreness increased from the pretest to the posttest for both conditions, hence attesting muscle inflammation [80]. The quadriceps was sorer than erector spinae, hamstring and adductors muscles. This is congruent with the nature of the movement and the imposed tempo. Compared to the low-bar squat, the back squat places a greater demand on quadriceps muscles due to a more upright trunk position [81–83]. The tempo required participants to control the eccentric phase and pause in a deep squat position. This unusual focus on eccentric and isometric phases elicits greater muscle damage [84, 85]. The between-conditions difference in the adductors DOMS was greater than that recorded for the erector spinae. Erector spinae muscles are primarily stabilizers during back squats [86]. Participants were trained and familiarized with the back squat, and their muscles accustomed to the isometric contraction regimen. The difference in adductor scores between conditions can be explained by the role of these muscles during back squat repetitions [87]. During the concentric phase of a back squat, hip adduction and internal rotation assist the hip extension [88]. Hence, the deleterious effects of SMR on quadriceps strength may have exacerbated the demands for adductors contraction. Muscle soreness was, irrespective of the muscle group, 0.67 points (66.3%) higher from pre-test to post-test under the control condition compared to SMR. As for reduced muscle swelling, this may primarily originate from the reduced total workload and concentric power under the SMR condition.

One might object that similar to SMR post-workout [15, 89], embedded SMR with a roller massager could have preventive effects on muscle damage.

Psychobiological frameworks conceptualize perceived fatigability as an increase in perceived exertion corollary of psychophysiological changes elicited by exercise [59, 90, 91]. Performance fatigability is typically associated with a decline in performance along with effort repetition [59, 92–94]. Concentric power decreased along with sets repetition. There was also an increase in perceived exertion across sets. For both variables, the slopes were similar between experimental conditions, attesting a decline of 23.02 W (-5.1%) in concentric power and an increase in perceived exertion of 0.84 arbitrary units (+3.6%) across sets. These findings demonstrate that SMR during the inter-set recovery periods of the resistance training protocol had no effect on the behavioral and psychophysical markers. It has previously been shown that SMR reduced muscle soreness resulting from exercise-induced muscle damage [15, 71, 80, 89, 95]. We measured reduced muscle soreness up to 120 h after completion of the resistance training protocol under the SMR condition. But this cannot be attributed to beneficial effects of SMR on exercise-induced muscle damage since, again, this could result from reduced total workload and concentric power under the SMR condition.

As with all research, several methodological limitations should be addressed. First, the investigation involved a limited small sample size, presumably due to restrictive inclusion criteria. The present pilot trial should be considered a very pilot trial, which precludes generalization of the findings to a larger population of athletes using SMR with a roller massage on a regular basis. Second, we did not implement objective monitoring of the pressures exerted by athletes on their soft tissues when performing SMR with the foam roller. This was primarily due to feasibility reasons. Some authors used a force platform to index pressures applied using foam rolling (with the foam roller on the floor) relative the individual body weight, but could not quantify the contact surface indicative of pressures relative a given pressure area [56, 96]. Such methods would not be applicable to SMR with a roller massager. By contrast, a subjective pain threshold represents a well-accepted methodological standard to control the levels of pressures administered during foam rolling or roller massager SMR [6, 8, 9, 54]. A subjective pain threshold corresponds actual practice contexts where athletes usually adjust their massaging actions based on their perceived sensations. The difference in total workload achieved by athletes between conditions prevents attributing the reduction of muscle swelling and soreness up to 120 h after the resistance training protocol to a benefit of SMR, since this occurred at the expense of the training intensity. Future studies should pay particular attention to controlling the total workload between conditions if their aim is to investigate the effects of embedded SMR post-workout, which was not the primary aim of the present design.

Present findings confirm the deleterious effect of embedded SMR with a roller massager on training intensity in a resistance training protocol [6, 9]. Embedded SMR with a roller massager was characterized by reduced total workload and increased fatigability, indexed from objective and subjective marker. The mechanisms underlying the negative effects of embedded SMR remain insufficiently understood [6, 8, 9, 20]. It is suggested that embedded SMR generates additional cognitive and physical demands compared to passive recovery which are detrimental to sort-term performances [8, 97], and that pressures elicit micro-damages to muscles that alter their contractile properties. Examining the effect of embedded SMR routines when administered by a physiotherapist, thus alleviating attentional and energetic costs associated with the regulation of SMR movements, would contribute to better understand optimal practice guidelines. This would have practical relevance for both coaches and athletes. Eventually, the SMR to passive recovery ratio was greater than that in the study by Monteiro et al. [9] (60 s of SMR, 4 min of recovery). It cannot be ruled out that the negative effects on performance

could be avoided below a certain threshold, e.g. below 0.5. Overall, the present pilot trial brings further insights on the effects embedded SMR.

## Supporting information

**S1 File. Raw data file.** Raw data collected during the pilot trial.
(CSV)

**S2 File. CONSORT checklist.** CONSORT checklist (extension for pilot trials).
(DOC)

**S3 File. CONSORT checklist.** CONSORT checklist short version (extension for pilot trials).
(DOC)

**S4 File. CONSORT flowchart.** CONSORT flowchart of the experimental procedures implemented for the pilot trial.
(DOCX)

**S5 File. Participants' information sheet (French version).** Original information document provided to the participants (French version).
(DOCX)

**S6 File. Participants' information sheet (English translation).** Original information document provided to the participants (English translation).
(DOCX)

## Author Contributions

**Conceptualization:** Yann Kerautret, Aymeric Guillot, Franck Di Rienzo.

**Data curation:** Yann Kerautret.

**Formal analysis:** Franck Di Rienzo.

**Investigation:** Yann Kerautret.

**Methodology:** Yann Kerautret.

**Project administration:** Yann Kerautret, Aymeric Guillot.

**Software:** Franck Di Rienzo.

**Supervision:** Aymeric Guillot, Franck Di Rienzo.

**Validation:** Aymeric Guillot, Franck Di Rienzo.

**Writing – original draft:** Yann Kerautret, Franck Di Rienzo.

**Writing – review & editing:** Aymeric Guillot, Franck Di Rienzo.

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
