## [Decision Letter · Decision Letter 0]

4 Jan 2021

PONE-D-20-34699

Evaluating the effects of embedded foam rolling practice on strength performance

PLOS ONE

Dear Dr. Di Rienzo,

Thank you for submitting your manuscript to PLOS ONE. After careful consideration, we feel that it has merit but does not fully meet PLOS ONE’s publication criteria as it currently stands. Therefore, we invite you to submit a revised version of the manuscript that addresses the points raised during the review process.

Please find attached three very informative, in-depth reviews to your submitted manuscript. I encourage you to be especially mindful of comments pertaining to the structure and gist of the introduction, the conceptual groundings of your study (regarding muscle damage, fatigue, metabolic aspects), and the methodology (sampling approach, familiarization, inter-individual differences). I agree wholeheartidly with comments regarding inaccurate wording, which are easily fixed and will greatly improve the message. Technical comments likewise.

Regarding Reviewer 1's comments about research vs clinical trial type article, I understand why this important distinction has to be made. I would like you to formulate a response, but since in the sports sciences we are merely following trends that already exist in the field, and not per se evaluating the efficacy of a medical device here (which other competent authorities have done), I am willing to treat this as a research article. That's an interesting grey area though.

We look forward to receiving your revised manuscript.

Kind regards,

Hugo A. Kerhervé, Ph.D.

Academic Editor

PLOS ONE

Journal Requirements:

2. Please amend either the title on the online submission form (via Edit Submission) or the title in the manuscript so that they are identical.

We note that one or more of the authors are employed by a commercial company: Capsix Robotics.

(2) Please also provide an updated Competing Interests Statement declaring this commercial affiliation along with any other relevant declarations relating to employment, consultancy, patents, products in development, or marketed products, etc.  

Please respond by return email with an updated Funding Statement and Competing Interests Statement and we will change the online submission form on your behalf.

Reviewers' comments:

Reviewer's Responses to Questions

**Comments to the Author**

1. Is the manuscript technically sound, and do the data support the conclusions?

Reviewer #1: Yes

Reviewer #2: Yes

Reviewer #3: Partly

2. Has the statistical analysis been performed appropriately and rigorously? 

Reviewer #1: No

Reviewer #2: Yes

Reviewer #3: Yes

3. Have the authors made all data underlying the findings in their manuscript fully available?

Reviewer #1: Yes

Reviewer #2: Yes

Reviewer #3: Yes

4. Is the manuscript presented in an intelligible fashion and written in standard English?

Reviewer #1: Yes

Reviewer #2: Yes

Reviewer #3: Yes

5. Review Comments to the Author

Reviewer #1: The manuscript presents an analyses of a randomized cross-over trial, aimed at examining the intervention effects of a foam rolling intervention. The study was IRB approved, however, I couldn't find the appropriate registration in clinicaltrials.gov (with a specific number) or within some other entity. If I am correct, it's not mentioned in the manuscript. The design looks adequate, however, I still have some questions on the design aspects of the study.

1. Sample size statement: With only n = 14 participants, I would be interested to know in advance what power one can expect in the analyses wrt. some pragmatic effect size. The writeup states: "Fourteen CrossFit athletes volunteered to participate...". You cannot conduct and analyze a clinical trial, unless it is sufficiently powered. Recruitment is expected to continue until the desired sample size is reached within a pragmatic time-frame. Else, it can be called a simple "pilot trial" designed to evaluate some initial effectiveness of an intervention. Again, it's not clear from the writeup. The authors should present a section on power/sample size considerations. There maybe R packages available for conducting power analyses under cross-over designs.

2. Writeup. The study has been submitted as a Research Article, and not a clinical trial. I am wondering, if the authors can refer clearly to the main clinical trial paper (already?) published from the randomized trial. Now, if that has not happened, the current writeup needs a thorough change, reflecting the CONSORT guidelines for submission of a clinical trial and it's analyses, and submitted as a Clinical Trial.

3. Randomization. Block randomization was used. What was the block size?

4. Statistical Analysis: Linear mixed effects, via R nlme, was used. How can one guarantee the Gaussian (Normal) behavior of the response variable? Was it checked?

5. Conclusions/Discussions: With n = 14, the study results can at best be reported as a very pilot study! This needs to be clearly mentioned.

Reviewer #2: I happen to have reviewed this manuscript before and I have to say this is a much improved version. I have a few minor comments regarding interpretation and language use to hopefully improve accuracy and readability of the manuscript – these can be found below:

Introduction:

It is, to some extent, unclear what the first two paragraphs of introduction are trying to achieve. They would read completely fine if this was a literature review. However, here, you are trying to introduce your experiment. It is thus unclear how the previous work presented in the first two paragraphs links to the present work – this becomes somewhat clearer later on in the introduction; but for readability sake, it would be much better if the reader didn’t have to make such connection themselves (in fact, a reader that is more naïve to the literature might struggle to make the connection).

Line 45: consisting ‘of’.

Line 52: ‘has been’ associated.

Line 54: static stretching should be singular (currently written as ‘statics’).

Line 56: ‘might participate’ is a strange term, please revise. I’m not entirely sure what you are trying to say in this sentence, otherwise I would suggest the alternative. Also, ‘athletic performance’ should be singular.

Line 57: Do you mean ‘repeated sprint performance’?

Line 63: ‘have been hypothesised’.

Line 63-66: How is this relevant to your investigation? Again, see my major comments about the Introduction, but this is a clear example of that.

Line 74: Do you mean ‘Monteiro et al.’?

Methods:

Line 138: I believe you have forgotten to translate ‘March’ from French into English.

Results:

Line 224: Did this differ statistically between groups?

Line 229: I believe ‘were greater’ is missing.

Line 254: Have you calculated measurement error for thigh circumference? Given the difference is quite small, it is important to establish a measurement error so the meaningful effect can be established (see some work by Monteiro et al. that showed the lack of meaningful difference even when the difference was statistically significant).

Line 268: Please reconsider the expression – ‘there WAS a significant interaction’ is much more appropriate (there is no such thing as an effect of an interaction when reporting results of the null-hypothesis significance testing).

Line 272: How can you be sure it approached significance and was not moving away from it? Please rephrase.

Discussion:

Line 302: consisted ‘of’.

Lines 305-306: You could provide the percentage change values in the brackets in the previous sentence; currently you are merely repeating the same information (with the exception of the quantification of a decrease) from the previous sentence, making text redundant and repetitive.

Line 320-321: I suggest this sentence be rephased. The expression to ‘compound the physical fatigue elicited by the resistance training protocol’ is unclear. Do you mean ‘could facilitate fatigability elicited by the resistance training protocol?’. Note that I also used the term ‘fatigability’ rather than ‘fatigue’ – the former describes an objective decline in a measure of muscle function during a discrete task, whereas the latter is a perception or feeling of exhaustion (see Enoka & Duchateau 2016 Med Sci, for clear distinction in terminology).

Line 321-323: There is no evidence for such a claim. In fact, research suggests that pressure far outside the physiological threshold are required to induce mechanical changes to tissue with foam rolling (see Chaudhry et al. 2008 J Am Osteopath Assoc).

Line 339-340: The baseline VAS scores being zero more likely indicates the lack of muscle damage/complete recovery than the lack of ‘residual muscle fatigue’. “Muscle fatigue” (or more appropriately termed ‘fatigability’) is a transient reduction in muscle force output that recovers withing minutes/hours of exercise. If the reduction in muscle force is persistent for days, it is typically classed as muscle damage (see Carroll et al. 2017, J Appl Physiol).

Line 371: ‘Physical’ seems a strange adjective for fatigue. Fatigue in its true sense of a word is a psychophysical phenomenon and one cannot necessarily be distinguished from the other (again, see Duchateau & Enoka 2016).

Line 371: It ‘has previously been shown’.

Line 377-378: The role of muscle damage in muscle hypertrophy is a debatable mechanism; e.g. mechanical signalling and metabolic stress are much more prominent triggers of the muscle remodelling process (see Wackerhage et al. 2019, J Appl Physiol).

The conclusion of the study seems rather abrupt. It would be prudent to briefly summarise the main findings (in a sentence or two) and infer meaning from it (e.g. functional consequence).

Reviewer #3: Below are some comments:

In Introduction more attention should be paid on explaining why foam rolling could influence on muscle performance in strength training as you hypothesized that embedded foam rolling with a

roller massager would reduce the strength performance.

Line 46: Author’s wrote: „(…)Foam rolling can be administered using

a foam roller, a roller massager, sticks or balls of varying sizes and densities (1,3,4)”. If you are writing foam rolling it determines specific tool known as foam roller. Of course you can use different equipment like you’ve mentioned but it wouldn’t be foam rolling. I think you meant Self Myofascial Release technique.

Line 51 and later in the text: please provide next numbers order

I don’t know why you compare male and females in the same group, as the training effects, somatic build, recovery capacity and many other factor significantly differ both sexes?

Could the authors add the sample size?

The description of the studied group lacks information on the sampling methodology. The authors do not state whether the selection was random or different, there is also no information about the sampling frame, as well as the method of reaching particular groups or participants - please add this information. It also needs to be clarified in the paragraph regarding study population how the participants were identified. Are the participants representative for a larger group of CrossFit Athletes or is it a very specific group? The authors must also consider how the recruitment process of the participants influence the results.

Lines 116-122: still it’s unclear how deep was the squat. wciąż jest niejasne jak określano głębokość przysiadu. Although it was constant, it does not follow from the description of whether it is a full, maximum deep squat or to a specific angle in the knee joint.

Line 127: please provide information about roller density

Line 128-129: how the level of pressure was determined?

Line 130: What was rolling routine?

Line138: Typo Mars instead of March

If familiarization session and two experimental session separated from each other by 10 consecutive days and in between days participants were instructed to avoid intensive training but otherwise we know that few consecutive days without proper stimulation could effect in lowering strength performance (e.g. Lucas Duarte Tavares, Eduardo Oliveira de Souza, Carlos Ugrinowitsch, Gilberto Candido Laurentino, Hamilton Roschel, André Yui Aihara, Fabiano Nassar Cardoso & Valmor Tricoli (2017) Effects of different strength training frequencies during reduced training period on strength and muscle cross-sectional area, European Journal of Sport Science, 17:6, 665-672, DOI: 10.1080/17461391.2017.1298673). Please comment on that.

Line 186: 15 cm above the top of the patella – why not in the widest place, it’s puzzling when you tak into consideration differences between participants?

Line 302: ecological doesn’t seem to be good description for typical conditions

Lines 329-330: wrong reference style

Lines 332-335: improved muscle metabolic activity as well as blood circulation was proved in recent study in PLOS One, please see: Adamczyk J.G., Gryko K., Boguszewski D. Does the type of foam roller influence the recovery rate, thermal response and DOMS prevention? PLoS ONE 15(6): e0235195. https://doi.org/10.1371/journal.pone.0235195

6. PLOS authors have the option to publish the peer review history of their article (what does this mean?). If published, this will include your full peer review and any attached files.

Reviewer #1: No

Reviewer #2: No

Reviewer #3: No

---

## [Author Response · Author response to Decision Letter 0]

12 Feb 2021

Please see "response to reviewers" file enclosed.

---

## [Decision Letter · Decision Letter 1]

16 Feb 2021

PONE-D-20-34699R1

Evaluating the effects of embedded self-myofascial release practice on strength performance: a randomized crossover pilot trial

PLOS ONE

Dear Dr. Di Rienzo,

Thank you for submitting your manuscript to PLOS ONE. After careful consideration, we feel that it has merit but does not fully meet PLOS ONE’s publication criteria as it currently stands. Therefore, we invite you to submit a revised version of the manuscript that addresses the points raised during the review process. You'll find minor points indicated below.

We look forward to receiving your revised manuscript.

Kind regards,

Hugo A. Kerhervé, Ph.D.

Academic Editor

PLOS ONE

Reviewers' comments:

Reviewer's Responses to Questions

**Comments to the Author**

1. If the authors have adequately addressed your comments raised in a previous round of review and you feel that this manuscript is now acceptable for publication, you may indicate that here to bypass the “Comments to the Author” section, enter your conflict of interest statement in the “Confidential to Editor” section, and submit your "Accept" recommendation.

Reviewer #1: All comments have been addressed

Reviewer #2: (No Response)

Reviewer #3: All comments have been addressed

2. Is the manuscript technically sound, and do the data support the conclusions?

Reviewer #1: (No Response)

Reviewer #2: Yes

Reviewer #3: Yes

3. Has the statistical analysis been performed appropriately and rigorously? 

Reviewer #1: (No Response)

Reviewer #2: Yes

Reviewer #3: Yes

4. Have the authors made all data underlying the findings in their manuscript fully available?

Reviewer #1: (No Response)

Reviewer #2: Yes

Reviewer #3: Yes

5. Is the manuscript presented in an intelligible fashion and written in standard English?

Reviewer #1: (No Response)

Reviewer #2: Yes

Reviewer #3: Yes

6. Review Comments to the Author

Reviewer #1: (No Response)

Reviewer #2: The authors are thanked for responding to my comments. I have a few minor comments that I believe would improve clarity.

Abstract – line 22: Self-myofascial release is a misleading/erroneous term as it is questionable whether they actually “release” fascia (see, for example, Behm & Wilke, 2019, Sports Med). Please rephrase to a more appropriate term throughout the manuscript.

Introduction:

Line 46: Consists ‘of’

Line 73: ‘has been shown’

Line 79: The notion of ‘compounding fatigue’ which I have highlighted as problematic in my previous review remains in this section. Please revise.

Discussion:

Line 335-338: According to the definitions proposed by WHO and adopted by the American College of Sports Medicine you would not have been controlling for the ‘sex’ rather than ‘gender’ effect. Particularly as you proceed to qualify it as ‘male’ and ‘female’, which are nouns associated with sex, rather than gender.

Line 365: You state “however, their [pressure] short-term effects on performance appeared here detrimental.” It is difficult to attribute this to pressure alone or rather disassociate pressure from other factors. For example, when pressure of roller massaging is controlled, the effects seem rather similar once a certain, very low baseline threshold is exceeded (Grabow et al. 2018, J Strength Cond Res).

Reviewer #3: Thank you for considering my comments. After improvements it's muc more clear. I think manuscript in current form can be published.

7. PLOS authors have the option to publish the peer review history of their article (what does this mean?). If published, this will include your full peer review and any attached files.

Reviewer #1: No

Reviewer #2: No

Reviewer #3: No

---

## [Author Response · Author response to Decision Letter 1]

17 Feb 2021

Please find the response to reviewer file enclosed.

---

## [Decision Letter · Decision Letter 2]

18 Feb 2021

Evaluating the effects of embedded self-massage practice on strength performance: a randomized crossover pilot trial

PONE-D-20-34699R2

Dear Dr. Di Rienzo,

We’re pleased to inform you that your manuscript has been judged scientifically suitable for publication and will be formally accepted for publication once it meets all outstanding technical requirements. Please be mindful of correcting some technical elements at the proofing stage (some spacing inconsistencies, unit "yr" missing line 93).

Kind regards,

Hugo A. Kerhervé, Ph.D.

Academic Editor

PLOS ONE

Reviewers' comments:

Reviewer's Responses to Questions

**Comments to the Author**

1. If the authors have adequately addressed your comments raised in a previous round of review and you feel that this manuscript is now acceptable for publication, you may indicate that here to bypass the “Comments to the Author” section, enter your conflict of interest statement in the “Confidential to Editor” section, and submit your "Accept" recommendation.

Reviewer #2: All comments have been addressed

2. Is the manuscript technically sound, and do the data support the conclusions?

Reviewer #2: Yes

3. Has the statistical analysis been performed appropriately and rigorously? 

Reviewer #2: Yes

4. Have the authors made all data underlying the findings in their manuscript fully available?

Reviewer #2: Yes

5. Is the manuscript presented in an intelligible fashion and written in standard English?

Reviewer #2: Yes

6. Review Comments to the Author

Reviewer #2: I thank the authors for taking my suggestions on board and incorporating them into the manuscript. I have no further comments.

7. PLOS authors have the option to publish the peer review history of their article (what does this mean?). If published, this will include your full peer review and any attached files.

Reviewer #2: No

---

## [Editor Report · Acceptance letter]

22 Feb 2021

PONE-D-20-34699R2 

Evaluating the effects of embedded self-massage practice on strength performance: a randomized crossover pilot trial 

Dear Dr. Di Rienzo:

I'm pleased to inform you that your manuscript has been deemed suitable for publication in PLOS ONE. Congratulations! Your manuscript is now with our production department. 

Kind regards, 

on behalf of

Dr. Hugo A. Kerhervé 

Academic Editor

PLOS ONE